# ESBL-Positive Enterobacteriaceae from Dogs of Santiago and Boa Vista Islands, Cape Verde: A Public Health Concern

**DOI:** 10.3390/antibiotics12030447

**Published:** 2023-02-23

**Authors:** Alice Matos, Eva Cunha, Lara Baptista, Luís Tavares, Manuela Oliveira

**Affiliations:** 1Veterinários Sem Fronteiras, Av. da Universidade Técnica, 1300-477 Lisboa, Portugal; 2CIISA—Centre for Interdisciplinary Research in Animal Health, Faculty of Veterinary Medicine, University of Lisbon, Av. da Universidade Técnica, 1300-477 Lisboa, Portugal; 3Associate Laboratory for Animal and Veterinary Sciences (AL4AnimalS), 1300-477 Lisboa, Portugal; 4Bons Amigos, Achada Grande Trás, Cidade da Praia 7603, Santiago Island, Cape Verde

**Keywords:** dogs, Cape Verde, antimicrobial resistance, ESBL, public health

## Abstract

Antimicrobial resistance is a public health threat with an increasing expression in low- and middle-income countries such as Cape Verde. In this country, there is an overpopulation of dogs, which may facilitate the spread of resistant bacteria, including extended-spectrum β-lactamase (ESBL)-producing Enterobacteriaceae. To clarify the role of dogs as reservoirs for the dissemination of this bacterial group, 100 rectal swab samples were collected from confined (n = 50) and non-confined (n = 50) dogs in Santiago and Boa Vista Islands, Cape Verde. These were analyzed using conventional bacteriological techniques for the detection of ESBL-producing Enterobacteriaceae and characterization of their pathogenic and resistance profiles. Twenty-nine samples displayed ESBL-positive bacteria, from which 48 ESBL-producing isolates were obtained and mostly identified as *Escherichia coli*. Multiple antimicrobial resistance indexes ranged from 0.18 to 0.70 and half of the isolates were classified as multidrug-resistant. Isolates were capable of producing relevant virulence factors, including biofilm, showing virulence indexes between 0.29 and 0.71. As such, dogs in Cape Verde may act as reservoirs of resistant bacteria, including pathogenic and zoonotic species, representing a public health concern. Although further investigation is needed, this study proposes the periodical analysis of dogs’ fecal samples to monitor resistance dissemination in the country, in a One-Health perspective.

## 1. Introduction

According to the World Health Organization (WHO), antimicrobial resistance (AMR) is a global growing threat that compromises the achievement of Sustainable Development Goals (SDGs) by endangering public health, food security, and economic growth [1]. Accordingly, in 2017, the WHO published a list of priority antimicrobial-resistant bacteria to guide research, in which extended spectrum β-lactamases (ESBL)-positive Enterobacteriaceae are included in the critical priority group [2]. They are also classified as a serious threat by the Centers for Disease Control and Prevention [3].

According to Bush–Jacoby updated functional classification, ESBL enzymes belong to group 2be and are associated with the resistance of bacteria to penicillins and cephalosporins as well as to their capability to hydrolyze one or more oxyimino-beta-lactams, such as cefotaxime, ceftazidime, or aztreonam [4]. ESBL enzymes are more or less effectively hydrolyzed by lactamase inhibitors, depending on the specific enzyme—TEM and SHV are better hydrolyzed by clavulanic acid while CTX-M is more successfully inhibited by tazobactam [4]. Although some ESBL isolates are susceptible to third generation cephalosporins in vitro, treatment with these antimicrobials may result in clinical failure [5], representing a concerning decrease in treatment options for infections caused by ESBL-producing Enterobacteriaceae.

Even though infectious diseases are an important cause of mortality in Cape Verde [6,7], the country is still lacking resources for infection control [8]. Furthermore, due to economic constrains, the use of antimicrobials is restricted to a few broad-spectrum compounds, and is mostly based on empirical prescription according to the patient’s clinical situation [6,8]. Moreover, there is under-reporting of clinical cases, which makes it difficult to assess the country’s real epidemiological situation, as well as the actual burden of disease [9] and the level of antimicrobial resistance. A five-year retrospective study of two hospitals in Cape Verde showed an alarming rise in resistant bacteria, namely of ESBL-producing strains [6]; however, information about this problem in Cape Verde is still scarce, especially in the veterinary setting.

Although bacteria are capable of developing a resistance profile naturally [10], this phenomenon has been accelerated by the overuse and misuse of antimicrobials, which may lead to selective pressure and acquired resistance [11]. As a consequence of the increasing reports of community-acquired resistant bacteria in humans [12,13,14], in addition to factors associated with hospital facilities and antimicrobial consumption, more attention has been given to different AMR sources and drivers, including overpopulation, poor sanitation, and lack of an efficient sewage-disposal system, which are phenomena observed in low- and middle-income countries (LMICs) [15,16,17], such as Cape Verde [18]. In addition, recent studies indicated the possible linkage between resistant bacteria of pets and their owners, or even unrelated humans. In fact, some authors demonstrated the co-carriage of antimicrobial-resistant bacteria between these hosts [19,20,21,22], while others showed that pathogenic bacteria harboring the same resistant genes, including *bla*_CTX-M_ and *bla*_TEM,_ are not species-specific, and therefore similar strains are capable of causing infection in both humans and companion animals [23,24,25,26]. Furthermore, dog feces are largely believed to be responsible for the transfer of pathogenic bacteria [27], as well as of resistant strains and genes [28,29,30] from one host to another. Accordingly, the fecal–oral route has been indicated as the main route for transmission of resistant pathogens such as those belonging to Enterobacteriaceae at the community level [15,31], which is sustained by the lack of hygiene practices [32].

Although official records of dog population numbers are lacking in Cape Verde, stray dogs are known to be a big concern on almost every island of the archipelago [33], especially in urban areas where food and other essential resources are more accessible [34]. These dogs live in close contact with humans and with hospital and municipal residues and wastewater, which can be high-risk contamination sources of antimicrobials [17,35]. As such, these animals are prone to develop and carry resistant bacteria, which can be disseminated through their fecal material to the community [27,28,29,30].

More studies on antimicrobial resistance dissemination need to be performed in LMICs, focusing on different AMR sources and drivers, including on companion animals. As such, this study aimed to evaluate the presence of ESBL-positive Enterobacteriaceae in dog feces in Cape Verde and to identify possible associations between ESBL-production and dogs’ lifestyle characteristics.

## 2. Results

Most of the sampled animals were female dogs (56%), between one and five years old (75%) (Appendix A). None of the sampled animals showed evident gastrointestinal signs of disease. Data regarding the spay/neuter status, the presence of external parasites, and the body-condition score of animals sampled at two different locations are highlighted on Table 1.

From the 100 samples collected, 66 gave positive cultures in ChromID^®^ESBL agar, resulting in 114 isolates. Despite that, after isolate identification and ESBL-production confirmation testing, only 48 isolates, obtained from 29 dogs, corresponded to ESBL-positive Enterobacteriaceae (Appendix A). Eleven of these dogs displayed more than one positive isolate. These were identified as *Escherichia coli* (n = 29; 60.42%), *Klebsiella pneumoniae* (n = 4; 8.33%), *Morganella morganii* (n = 4; 8.33%), *Citrobacter freundii* (n = 3; 6.25%), *Proteus mirabilis* (n = 3; 6.25%), *Escherichia vulneris* (n = 3; 6.25%), *Enterobacter cloacae* (n = 1; 2.08%), and *Proteus* sp. (n = 1; 2.08%).

### 2.1. Characterization of the Isolates’ Antimicrobial-Resistance Profile

Besides being resistant or intermediately resistant (n = 48; 100%) to ampicillin, cefotaxime, and ceftaroline, as observed after modified double-disk synergy testing, all ESBL-positive isolates were also frequently resistant or intermediately resistant to enrofloxacin (n = 38; 79%), doxycycline (n = 30; 32%), and amoxicillin/clavulanate (n = 24; 50%). On the other hand, all isolates were susceptible to gentamicin (120 µg) and meropenem (Table 2). Despite that, 94% (n = 45) of the isolates were resistant or intermediately resistant to at least one antimicrobial class besides β-lactams and excluding antimicrobial compounds for which intrinsic resistance is described for Enterobacteriaceae. Moreover, 50% (n = 24) of the isolates were classified as multidrug-resistant (MDR) considering their non-susceptibility to three or more antimicrobial classes [36].

The multiple antimicrobial resistance (MAR) indexes ranged between 0.18 and 0.70 (Appendix A), with only one isolate displaying an index below 0.20. The highest mean value was displayed by the *K. pneumoniae* isolates (MAR index mean value = 0.65), followed by *C. freundii* (MAR index mean value = 0.59), *M. morganii* (MAR index mean value = 0.57), *E. coli* (MAR index mean value = 0.42), *P. mirabilis* (MAR index mean value = 0.41), *Proteus* sp. (MAR index value = 0.33), *E. cloacae* (MAR index value = 0.33), and *E. vulneris* (MAR index mean value = 0.27).

No significant differences between MAR index values from isolates obtained from dogs of different groups (dogs from different locations, dogs with different confinement status, dogs with different body condition score, and dogs with and without external parasites) were detected (*p* > 0.05).

### 2.2. Characterization of the Isolates’ Virulence Profile

Most isolates were able to produce lipase (n = 44; 91.67%), hemolysins (n = 37; 77.08%), and biofilm (n = 27; 56.25%), but other virulence factors were not as commonly expressed by the isolates, including DNase (n = 18; 37.50%), gelatinase (n = 12; 25.00%), and protease (n = 11; 22.92%) (Appendix A). None of the isolates was positive for lecithinase production.

The *Proteus* sp. isolate showed the highest virulence index (V. index value = 0.71), followed by *P. mirabilis* (V. index mean value = 0.57), and *K. pneumoniae* (V. index mean value = 0.54). Lower V. indexes were displayed by *Citrobacter freundii* (V. index mean value = 0.48), *M. morganii* (V. index mean value = 0.46), *E. coli* (V. index mean value = 0.42), *E. cloacae* (V. index value = 0.29), and *E. vulneris* (V. index mean value = 0.29). Virulence indexes varied between 0.29 and 0.71.

No significant differences in virulence index values from isolates originated from dogs from different groups (dogs with different confinement status, dogs with different body condition score, and dogs with and without external parasites) were detected (*p* > 0.05), except for isolates from different sampling locations (*p* = 0.006).

Correlation between isolates’ MAR index and V. index was not statistically significant (*p* > 0.05). MAR index values showed no significant differences between biofilm-producing isolates (MAR index mean value = 0.47) and non-producers (MAR index mean value = 0.41) (*p* > 0.05). However, biofilm production was found to be significantly related to MDR profile (*p* = 0.042; OR = 3.4; 95% CI 1.03–11.26).

## 3. Discussion

To the best of the authors’ knowledge this is the first study on ESBL-producing Enterobacteriaceae isolated from fecal samples of dogs from Cape Verde, providing not only relevant information on the intestinal microbiota of these dogs, but also exposing possible risks for both animal and public health.

Samples were obtained using rectal swabs with Amies transport medium (VWR™, Leuven, Belgium), which is described as effective and reliable for fecal bacterial isolation [37,38]. The isolates were kept refrigerated at 4 °C, allowing a delayed analysis and a long-distance transportation [38,39].

Detection of ESBL-producing Enterobacteriaceae was done using a selective chromogenic agar (ChromID^®^ESBL agar). In agreement with our results, this selective and chromogenic medium is considered a sensitive, but not very specific, method for rapid detection and presumptive identification of ESBL-positive isolates [40,41,42], which emphasizes the importance of doing a confirmatory test for a correct detection of this resistance mechanism. Considering that other authors reported false-negative results in the confirmatory test recommended by CLSI [43] associated with AmpC β-lactamase production, in the present study the modified double-disk synergy test was performed instead, using also cefepime, which is stable to AmpC hydrolysis [44,45,46,47]. None of the isolates under study seemed to be co-producer of ESBL or AmpC.

The high-occurrence of ESBL-positive bacteria among the sampled dogs may be related to their potential contact with specific drivers and contaminated environments, such as urban residues and hospital wastewater [17,35]. Even though *p* values were above 0.05, the frequency of ESBL-positive samples was higher in dogs from Praia, in not-confined dogs, in dogs presenting external parasites, and in dogs with poor body condition. It is also important to notice that the chi-square test of dependency between ESBL and the external parasites and body condition variables displayed *p* values of 0.06, which points to the need to evaluate a larger animal population in future research. Additionally, none of the sampled animals showed gastrointestinal signs, which is in agreement with previous findings that healthy animals, without diarrhea, can also be ESBL carriers and act as reservoirs [26,48,49].

Among the confirmed ESBL-positive Enterobacteriaceae isolates, *E. coli* was the most prevalent species, as frequently reported in human and veterinary medicine and widely described as the cause of unsuccessful therapies in humans [26,50]. Accordingly, the identified bacterial species correspond to opportunistic pathogens responsible for intestinal and extraintestinal diseases, being associated with mild to severe infections in small-animal clinical practice [51,52], but also implied in a myriad of infections in humans, including urinary-tract infections, pneumonia, and septicemia, either hospital- or community-acquired [12,13,14,53,54]. 

Antimicrobial susceptibility profiles of the isolates under study are of special concern since the isolates tested were resistant to antimicrobials that are frequently used in both human and veterinary medicine [55,56], with results similar to those reported in human medicine.

According to CLSI guidelines [57], despite their susceptibility profiles, all isolates must be declared resistant to penicillins, cephalosporins and aztreonam once identified as ESBL-producers. In addition, 50% of the isolates showed no susceptibility to amoxicillin/clavulanate, demonstrating that ESBL-producing bacteria can develop resistance to more molecules, as previously reported [58,59]. Another relevant result was the lack of susceptibility to fluoroquinolones, namely to enrofloxacin, but also to ciprofloxacin. High occurrence of fluoroquinolones resistance among ESBL-positive isolates has already been reported in humans by Kantele et al. [60] and in dogs Johansson et al. [61], who suggested that this phenomena can be due to conjugative spread of *bla* genes by plasmid transfer.

Doxycycline was also included in this study because it is widely used in Cape Verde to treat a variety of bacterial infections in both veterinary and human medicine [62,63,64]. Resistance to this antimicrobial has already been reported among ESBL-positive Enterobacteriaceae species isolated from humans and dogs [65,66], which is in agreement with this study’s results.

Nitrofurantoin is considered very effective against *E. coli* [67,68] and a reasonable alternative to fluoroquinolones and β-lactams in the treatment of UTIs in human medicine and small-animal clinical practice [67,69,70]. As expected, all *E. coli* isolates were susceptible to this antimicrobial, and only three *K. pneumoniae* isolates were resistant to nitrofurantoin.

In addition to the numerous reports of high-level gentamicin-resistant *Enterococcus faecum* isolates from both humans and companion animals [71,72,73,74], there are also few reports of high-level aminoglycoside resistance among Enterobacteriaceae [75,76]. In the present study, 40% of the tested isolates were not susceptible to low-content gentamicin disks (10 µg), but none were high-level gentamicin-resistant, which points to the importance of establishing proper doses regarding antimicrobial therapy.

Although there are increasing reports of carbapenem-resistant Enterobacteriaceae in veterinary and human medicine [77,78,79,80,81], all isolates were susceptible to meropenem, which seems to indicate that resistance to this last-resort antimicrobial is not a major problem in Cape Verde.

Genes encoding for resistance strategies can be horizontally transferred between bacteria, mainly via plasmids, thus enabling bacteria to display a wide range of mechanisms [60]. Conjugative plasmid-exchange might contribute to the high percentage (50%) of MDR ESBL isolates observed in this study, although to confirm this hypothesis it would be necessary to perform the molecular characterization of these isolates. 

Expression of virulence determinants by bacteria also influences clinical presentations and outcomes of associated infections, and plays a very important role in bacteria’s pathogenic potential by enabling the microorganism to invade the host and cause disease [82]. In the present study, most of the isolates showed capacity to produce lipases, hemolysins, and biofilms, which are surface-attached bacterial communities wrapped in an amorphous polysaccharide matrix that acts as a shield [83,84,85]. This ability enhances bacterial dissemination, confers survival advantages to host defenses, and is responsible for increased antimicrobial resistance through various mechanisms [83,86,87], which may explain why the isolates under study that were able to produce biofilms showed a statistically significant likelihood of being MDR. 

The present study demonstrated that fecal carriage of ESBL, MDR, and potentially pathogenic bacteria may occur among dogs from Cape Verde, independently of their confinement and health status. Considering that, in the locations where this study was performed, most of the dogs, including the confined ones, defecate in public streets that are not often cleaned, the carriage of resistant bacteria by these animals may represent a public and environmental health problem, as suggested previously [29,30,88,89]. In addition, several authors have suggested that some bacteria, including Enterobacteriaceae species found in the present study, can act as zoonotic agents, enabling the transfer of relevant antimicrobial resistance and virulence determinants between bacteria of dogs and humans [19,20,21,22,24,25,89,90]. Therefore this study represents the first step towards the understanding of the actual role of dogs as AMR reservoirs in Cape Verde. 

It is important to highlight that sample size depended on the clinical appointments in each rescue association during the time period available for sample collection, and not on the dog population size in each location, which is still uncertain, meaning the dataset has some limitations regarding representability. Another limitation of the present study is the lack of information regarding previous hospitalizations and therapies administered to the animals, which can influence the fecal carriage of ESBL at the time of the collection [91,92]. Possible miscommunication with owners about dog’s confinement status may also have had an impact on the lack of association between confinement status and ESBL-positive samples. Moreover, the available data do not provide information about the isolates genetic relatedness and epidemiology, and this should be considered in future studies.

## 4. Materials and Methods

### 4.1. Sampling Area

Sampling was performed between September and December 2021, in two different locations of Cape Verde (Figure 1): Bons Amigos Association, located in the municipality of Praia (14°55′15″ N, 23°30′30″ W), Santiago Island, and Nerina Association, located in Sal Rei (16°10’45” N, 22°55’ W), Boa Vista Island.

Praia is one of the nine municipalities of Santiago Island, with more than 140,000 inhabitants [93] sharing 258 km^2^ [94]. Praia has a high urbanization level (97.1%), one hospital and six public-health care centers. Boa Vista Island is 631 km^2^ but has only one municipality, Boa Vista, with a total of 12,613 residents corresponding to less than 3% of the national resident population [64]. Boa Vista has an urbanization level of 86.7%, and only one public-health care center. Animal density in both location is difficult to determine, and it was estimated that there were about 20,000 dogs in Praia and 2000 in Boa Vista [95].

### 4.2. Sample Collection

Samples were collected during clinical check-ups or during spay/neuter campaigns organized by the two associations. In each location, fifty dog fecal samples—half confined (C) and half not confined (NC)—were collected. Confined dogs are defined as those who lived in a household and were able to go out only with a dog leash, while non-confined dogs were free-roaming, independent of their ownership. No other inclusion criteria were considered.

After cleaning the perineal area using wipes to avoid contamination from the surrounding skin [96], samples were collected using a sterile swab (VWR™, Leuven, Belgium), inserted approximately 2 cm into the dog’s rectum [37] and gently rotated. Immediately after, the swab was placed in the respective tube containing Amies media (VWR™, Leuven, Belgium), and refrigerated at 4 °C until transport and further processing at the Laboratory of Microbiology and Immunology of the Faculty of Veterinary Medicine, University of Lisbon, Portugal.

Complementary animal data, including age (≤0.5 years; ≤1 years; ≤5 years; >5 years), sex (male/female), body condition score (1—severely underweight; 2—underweight; 3—ideal weight; 4—overweight; 5—obese), spay/neuter status (yes/no), presence of external parasites (yes/no), and major clinical signs, were also registered. It was not possible to record information regarding previous antimicrobial therapy or hospitalizations.

### 4.3. Isolation and Identification of ESBL-Positive Bacteria

Samples were inoculated on Brain–Heart Infusion (BHI) broth and incubated at 37 °C for 24 h. All suspensions were then inoculated on a selective chromogenic agar (chromID^®^ ESBL, bioMérieux, Linda-a-Velha, Portugal) and incubated at 37 °C for 24 h, to isolate the ESBL-positive bacteria. After incubation, the color of colonies was recorded, and bacteria were isolated in BHI agar before further testing.

Isolates were characterized considering their macroscopic morphology on ChromID^®^ESBL according to the manufacturer’s instructions (bioMérieux, Linda-a-Velha, Portugal), Gram staining, lactose-fermentation capability on MacConkey agar (Oxoid, Hampshire, UK), and oxidase production. Biochemical identification was performed by IMViC testing [97] or, in case of inconclusive result by IMViC, API 20E (bioMérieux, Linda-a-Velha, Portugal) galleries were used according to manufacturer’s instructions.

### 4.4. Evaluation of Isolates’ Antibiotic-Resistance Profile

To avoid false-negative results due to AmpC β-lactamase production, Enterobacteriaceae isolates were submitted to a modified double-disk synergy test to confirm ESBL production using four antimicrobials, namely two third-generation cephalosporins, cefotaxime (CTX, 30 µg) and ceftazidime (CAZ, 30 µg), a fourth-generation cephalosporin, cefepime (FEP, 30 µg), and amoxicillin/clavulanate (AMC, 30 µg). Antimicrobial disks were placed 20 mm apart, on the surface of a Mueller–Hinton agar plate previously inoculated with 0.5 MacFarland bacterial suspensions [44,45,46,47]. A positive result was recorded when, after 24 h incubation at 37 °C, a decreased susceptibility to the tested cephalosporins was detected together with a key-hole-shaped halo in the AMC disk, as a result of higher inhibition [45]. *Klebsiella pneumoniae* CFCT 7787 [98] and the reference strain *E. coli* ATCC^®^ 25922™ were used as positive and negative controls, respectively [46].

The disk diffusion method was performed to determine the isolates’ susceptibility profile, according to the Clinical Laboratory Standards Institute (CLSI). Eleven different antimicrobial compounds used in veterinary and human medicine, belonging to five different classes were tested: β-lactams, namely cephalosporins (third-generation—cefotaxime CTX, 30 µg and fifth-generation—ceftaroline CPT, 30 µg), penicillins (ampicillin AMP, 10 µg), penicillins in combination with a β-lactamase inhibitor (amoxicillin/clavulanate AMC, 30 µg), and carbapenems (meropenem MEM, 10 µg); aminoglycosides (gentamicin CN, 120 µg and 10 µg); fluoroquinolones (ciprofloxacin CIP, 5 µg, and enrofloxacin ENR 5 µg); nitrofurans (nitrofurantoin F, 100 µg); and tetracyclines (doxycycline DO, 30 µg). Antimicrobial disks were placed in the surface of a Mueller–Hinton agar plate previously inoculated with 0.5 MacFarland bacterial suspensions, which were incubated 24 h at 37 °C. Then, inhibition zones were measured and scored according to CLSI guidelines as susceptible, intermediate, or resistant [43,57]. The reference strain *E. coli* ATCC^®^25922™ was used for quality control, and a 10% replica was performed.

Multiple antimicrobial resistance (MAR) indexes were calculated as the quotient between the number of antimicrobials to which isolates were resistant and the number of antimicrobials tested [99].

### 4.5. Phenotypic Evaluation of Isolates’ Virulence Profile

Isolates’ virulence profile was also assessed for the ESBL-positive isolates by evaluating the production of certain enzymes related to the bacteria’s pathogenic potential. A 10% replica was performed.

Hemolysin expression was determined using Columbia agar plates supplemented with 5% sheep blood (bioMérieux, Linda-a-Velha, Portugal). DNase production ability was determined using DNase medium (Thermo Scientific™ Remel™, Porto Salvo, Portugal) supplemented with 0.01% toluidine blue O reagent, using *Staphylococcus aureus*, ATCC^®^ 25923™ and *E. coli*, ATCC^®^ 25922™ as positive and negative control, respectively. Lipase production was assessed using Spirit Blue (Difco™, BD Life Sciences, Vaud, Switzerland) agar added with a lipid source, olive oil, and Tween^®^80, using *Pseudomonas aeruginosa* ATCC^®^ 27853™ as positive control and a *P. aeruginosa* Z25.1 clinical isolate from a diabetic foot infection as negative control. Lecithinase activity was determined using tryptic soy agar supplemented with 10% egg yolk emulsion (VWR™, Leuven, Belgium), with *P. aeruginosa* ATCC^®^ 27853™ and *E. coli* ATCC^®^25922™ acting as positive and negative controls, respectively [99,100]. For evaluation of protease production, skim-milk powder (Oxoid, Hampshire, United Kingdom) supplemented with bacteriological agar (VWR™, Leuven, Belgium) was used, with *P. aeruginosa* ATCC^®^ 27853™ (positive) and *S. aureus* ATCC^®^ 29213™ (negative) being tested as controls [99]. Gelatinase activity was detected using Nutrient Gelatin Agar (Oxoid, Hampshire, United Kingdom), using *P. aeruginosa* Z25.1, a clinical isolate from a diabetic foot infection, and *E. coli* ATCC^®^ 25922™, as positive and negative controls, respectively [100]. Finally, biofilm formation was assessed using Congo red agar plates, composed of BHI broth (VWR™, Leuven, Belgium), bacteriological agar (VWR™, Leuven, Belgium), Red Congo reagent (Sigma-Aldrich, St. Louis, MO, USA), and sucrose, and *P. aeruginosa* ATCC^®^ 27853™ (positive) and *E. coli* ATCC^®^ 25922™ (negative) as controls. 

Virulence indices were calculated according to Fernandes et al. [99] as the quotient between the number of positive virulence factors expressed by an isolate and the number of virulence factors tested.

### 4.6. Statistical Analysis

Statistical analysis was performed using R commander (R^©^ v. 4.2.1 GUI 1.79).

The chi-square test was done to evaluate the dependency between ESBL production and possible animal-related predictors such as sampling location, confinement status, presence of external parasites, and body condition, and to analyze the association between biofilm production by the isolates under study and other virulence factors, as well as between biofilm production and the detection of a multidrug-resistance profile. Fisher’s exact test was used instead of chi-square when test assumptions were not met, and odds ratio was determined to measure these associations.

The Wilcoxon rank sum test was performed to test differences between MAR and virulence indexes of dogs from Praia and Boa Vista, confined and not confined dogs, with and without external parasites, and dogs with different body condition. The same test was used to assess possible differences between MAR index values of biofilm producers and non-producers. The Spearman’s correlation test was performed to analyze the correlation between the MAR index and virulence index.

Significant differences were calculated at 0.05 (two-tailed) levels of significance.

## 5. Conclusions

Our study revealed a high frequency of MDR and ESBL-producing isolates among dogs in two islands of the Cape Verde archipelago. This resistance phenotype was found in samples from different groups, suggesting its association with various risk factors, including population density and inefficient sewage and residues disposal, which should be explored in future studies.

Furthermore, the results of the present work will hopefully encourage further investigation on the role of companion animals as antimicrobial resistance reservoirs in Cape Verde, specifically by performing molecular characterization of the isolates and a public-health risk assessment. It could be beneficial to perform periodic analysis of dogs’ fecal samples with the aim of obtaining relevant information to track AMR tendencies, in an One-Health perspective, and it is urgent to start analyzing human clinical and non-clinical isolates to identify related trends.

Finally, dog-population-management initiatives, including reproduction control, community education campaigns and legislation, are essential to reduce the base problem of dog overpopulation in this country.

## Figures and Tables

**Figure 1 antibiotics-12-00447-f001:**
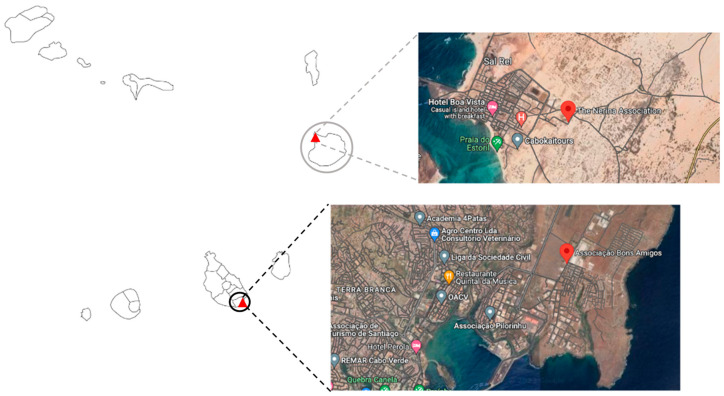
Map of Cape Verde and study area. The municipality of Praia is marked with a black circle, and Bons Amigos Association (Bons Amigos) with a red pin. The municipality of Boa Vista is marked with a grey circle, and Nerina Association (Nerina) with a red pin. The map was adapted from Vemaps^©^ (vemaps.com) and Google Maps (maps.google.com).

**Table 1 antibiotics-12-00447-t001:** Sampled animals’ data by sampling location.

	Spay/Neuter Status		Presence of External Parasites		Body Condition Score				
	YES	NO	YES	NO	1	2	3	4	5
BA (n = 50)	n = 0	n = 50 (100%)	n = 46 (92%)	n = 4 (8%)	n = 6 (12%)	n = 22 (44%)	n = 20 (40%)	n = 2 (4%)	n = 0
Ne (n = 50)	n = 35 (70%)	n = 15 (30%)	n = 12 (24%)	n = 38 (76%)	n = 1 (2%)	n = 12 (24%)	n = 34 (68%)	n = 3 (6%)	n = 0

Bons Amigos Association (BA); Nerina Association (Ne); number of isolates (n = x); percentage (%).

**Table 2 antibiotics-12-00447-t002:** Antimicrobial susceptibility profile of the isolates under study, regarding eleven antimicrobials belonging to five different classes.

Antimicrobial Class	Antimicrobial Compound	Concentration (µg)	Bacterial Isolates (n = x (%))		
			S	I	R
β-lactams	AMP	10	0	0	48 (100)
	CTX	30	0	1 (2)	47 (98)
	CPT	30	0	3 (6)	45 (94)
	AMC	30	24 (50)	9 (19)	15 (31)
	MEM	10	48 (100)	0	0
Fluroquinolones	CIP	5	26 (54)	9 (19)	13 (27)
	ENR	5	10 (21)	13 (27)	25 (52)
Tetracyclines	DO	30	18 (38)	4 (8)	26 (54)
Aminoglycosides	CN	10	29 (60)	1 (2)	18 (38)
	CN	120	48 (100)	0	0
Nitrofurans	F	100	41 (85)	0	7 (15)

Number of isolates (n = x); percentage (%); susceptible (S); intermediate (I); resistant (R); cefotaxime (CTX); ceftaroline (CPT); ampicillin (AMP); amoxicillin/clavulanate (AMC); gentamicin (CN); doxycycline (DO); enrofloxacin (ENR); ciprofloxacin (CIP); nitrofurantoin (F); meropenem (MEM). Note 1: *K. pneumoniae*, *E. cloacae*, *C. freundii*, and *M. morganii* have intrinsic resistance to ampicillin (AMP). Note 2: *E. cloacae*, *C. freundii*, and *M. morganii* have intrinsic resistance to amoxicillin/clavulanate (AMC). Note 3: *P. mirabilis* and *M. morganii* have intrinsic resistance to doxycycline (DO). Note 4: *P. mirabilis* and *M. morganii* have intrinsic resistance to nitrofurantoin (F).

## Data Availability

The datasets used and/or analyzed in the current study are available from the corresponding author upon reasonable request.

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
