# Peer review of "ESBL-Positive Enterobacteriaceae from Dogs of Santiago and Boa Vista Islands, Cape Verde: A Public Health Concern"

_antibiotics, 2023, doi:10.3390/antibiotics12030447_

Round 1

Reviewer 1 Report

I want to congratulate the authors for the impressive paper writing and very interesting subject. My concerns regarding the article are related to a few items.

1. The article has no conclusions. I would recommend including a conclusion section.

2. The discussion section is very theoretical, I would suggest comparing the antibiotic resistance of germs identified in dogs to ones identified in humans (with examples and compared percentage of resistance). The discussion section starts with "role as reservoirs for potentially resistant and pathogenic bacteria that can represent a risk for both animal and public health" (line 154-155). 

3. Data found in results is not interpreted in the discussion section. I suggest more comparison between own results and literature.

4. I suggest positioning Materials and Methods immediately after the Introduction section.

Author Response

Dear Reviewer,

We would like to thank you for the comments and the discussion points presented. We agree that they will improve the quality and understanding of the manuscript. Please see below the response to each of your question. Also, all changes performed in the revised version of the manuscript were highlighted in yellow.

  1. The article has no conclusions. I would recommend including a conclusion section.

As suggested, we included a conclusion section in Lines 404-419.

  1. The discussion section is very theoretical, I would suggest comparing the antibiotic resistance of germs identified in dogs to ones identified in humans (with examples and compared percentage of resistance). The discussion section starts with "role as reservoirs for potentially resistant and pathogenic bacteria that can represent a risk for both animal and public health" (line 154-155). 

Thank you for your comment. We have improved the discussion section with the inclusion of new references on antimicrobial resistance in humans and dogs as suggested (see lines 192-233).

  1. Data found in results is not interpreted in the discussion section. I suggest more comparison between own results and literature.

We understand your point. We have increased the discussion section and included new references as you may see in lines 183-191 and 214-238.

  1. I suggest positioning Materials and Methods immediately after the Introduction section.

We understand your remark, however the location of the material and methods in the end of the manuscript is a guideline of this journal, so we cannot change it.

Best regards,

Eva Cunha

Reviewer 2 Report

In the manuscript titled “ESBL-Positive Enterobacteriaceae from dogs of Santiago and Boa Vista Islands, Cape Verde: A Public Health concern,” the authors examined 100 rectal swabs from dogs to clarify their role as reservoirs for the dissemination of Extended Spectrum β-Lactamases (ESBL)-producing Enterobacteriaceae. The authors used traditional methods for identifying the isolates to the species level. To identify ESBL activity, the authors used chromogenic media and the Modified Double Disc Synergy Test. Also, the authors determined the phenotypic resistance profile for their isolates against 10 antimicrobials. Finally, the authors examined some phenotypic characters related to the virulence of their isolates.

Introduction:

 Please enrich the introduction with recent literature (the last five years).

L38-41: Please add more brief scientific details on the classification, families, and genetics of ESBLs.

Please add recent literature on the epidemiology of ESBL-producing bacteria in dogs and humans. 

Material and methods:

Although the estimated number of dogs in Praia is ten times more than Boa Vista, an equal sample size was collected from both locations. Did the authors follow any statistical formula for sample size calculation during their study design?

I think the resolution of Figure 1 is not enough. Please try to adjust it.

The authors collected some complementary animal data. Is there any data related to the dog owners among the confined group? For example, recent use of antibiotics, recent hospitalization, and current clinical signs.

  L271-272: “Biochemical identification was performed by IMViC testing [70] or API 20E” What did the authors mean? Are they used one of both tests? The methodology should be fixed for all tested samples to ensure accurate results.

Why did the authors not use molecular tools to identify their isolates, such as 16 s rRNA?

L276: “MDDST” Please add the full name “Modified Double Disc Synergy Test.”

L281: For phenotypic identification of ESBLs, the authors followed the methods described previously by Kaur et al. 2013, who used the CLSI guidelines to interpret their results. Why the authors used the EUCAST guidelines during this work?  (Ref. No 74).

L293: “aminoglycosides (gentamicin CN, 120 µg and 10 µg).” why the authors used gentamicin 120 µg? and how did the authors interpret the results of gentamicin sensitivity?

L297-299: “Then, inhibition zones were measured and scored according to CLSI guidelines as susceptible, intermediate, or resistant [48, 74]. If the authors used CLSI guidelines so, why they cited the EUCAST guidelines?

One of the most important mechanisms of antimicrobial resistance in Enterobacteriaceae is the enzymatic inactivation of penicillins and cephalosporins by means of plasmid-mediated extended-spectrum β-lactamases (ESBLs), such as the TEM-, SHV-, or cefotaxime (CTX)-M-group enzymes. Why do the authors use molecular tools to identify their isolates?

L304: “Phenotypic evaluation of isolates’ virulence profile,” the authors determine some phenotypic properties of their isolates. But they do not identify the pathotypes of isolated E. coli and whether the Klebsiella pneumoniae are hypervirulent strains. Molecular detection of such virulence genes is essential.

Results

Based on the MDDST, the authors identified 48 ESBL-producing isolates from 29 dogs. In total, how many samples revealed more than isolate? Please clarify this point in the result section.

In supplementary materials, 6 isolates were recovered from the sample (ID 5). Four of them were ESBL-positive E. coli. Are they all the same strain? I think Multilocus sequence typing (MLST) is required to differentiate these isolates.

Discussion:

Please add your study limitation and a final conclusion.  

Author Response

Dear Reviewer,

We would like to thank you for the comments and the discussion points presented. We agree that they will improve the quality and understanding of the manuscript. Please see below the response to each of your question. Also, all changes performed in the revised version of the manuscript were highlighted in yellow.

  1. Please enrich the introduction with recent literature (the last five years).

We understand your point and included recent literature (see references 19-22, 24-26, 29-31)

  1. L38-41: Please add more brief scientific details on the classification, families, and genetics of ESBLs.
  2. Please add recent literature on the epidemiology of ESBL-producing bacteria in dogs and humans. 

We have included new information regarding ESBL as suggested in the introduction section (see lines 39-45 and 61-77)

Material and methods:

  1. Although the estimated number of dogs in Praia is ten times more than Boa Vista, an equal sample size was collected from both locations. Did the authors follow any statistical formula for sample size calculation during their study design?

We understand your question. In fact, we would like to have proportional samples from both locations, however, that was not possible due to the number of animals that were present in the associations. This information was added as a limitation of the study in the discussion section (lines 260-263).

  1. I think the resolution of Figure 1 is not enough. Please try to adjust it.

Than you for your observation, we improved the resolution of the figure 1.

  1. The authors collected some complementary animal data. Is there any data related to the dog owners among the confined group? For example, recent use of antibiotics, recent hospitalization, and current clinical signs.

We understand your question, however, it was not possible to obtain information regarding previous antimicrobial therapy or hospitalizations of the animals, as we describe in the material and methods section (Lines 311-312). This is a very particular context, since animals’ owners have low resources to invest in animals’ care. This point was also added in the discussion section as a limitation of the study (please see lines 263-266).

L271-272: “Biochemical identification was performed by IMViC testing [70] or API 20E” What did the authors mean? Are they used one of both tests? The methodology should be fixed for all tested samples to ensure accurate results.~

We understand your question. We performed API20E in cases of inconclusive results on IMViC test (please see line 323).

  1. Why did the authors not use molecular tools to identify their isolates, such as 16 s rRNA?

We understand your comment. In fact, molecular tools are good methods to identify bacteria. However biochemical methods are quick and easy to perform techniques, and economic, being usually used in clinical bacteriology laboratories for identification of clinical isolates at the species level. We have long expertise and knowledge in using biochemical tests for bacteria identification from clinical samples (Approved Training Center by the European College of Veterinary Microbiology) and they are currently used in our laboratory to identify Enterobacteriaceae from animals’ origin. This was the reason why we used this methodology, besides the economical factor.

We were able to construct a bacterial collection and to evaluate for the first time the potential of dogs to act as reservoirs of antimicrobial resistance in Cape Verde, which was our main goal. This type of study in low income countries are scarce and very useful to the local authorities. For that reason and despite the molecular characterization of the isolates was not performed yet, the results obtained are very relevant to this population and context.  

  1. L276: “MDDST” Please add the full name “Modified Double Disc Synergy Test.”

Thank you for your indication. We have change it.

  1. L281: For phenotypic identification of ESBLs, the authors followed the methods described previously by Kaur et al. 2013, who used the CLSI guidelines to interpret their results. Why the authors used the EUCAST guidelines during this work?  (Ref. No 74).

We appreciate your question. We have used the CLSI guidelines to determine the susceptibility profile of the isolates, as described in the methods section, since CLSI has specific breakpoints for isolates of veterinary origin. We did not use CLSI for the modified double disk synergy test, since it does not have breakpoints for this test. As such, the EUCAST Guidelines were used to perform the modified double disk synergy test.

  1. L293: “aminoglycosides (gentamicin CN, 120 µg and 10 µg).” why the authors used gentamicin 120 µg? and how did the authors interpret the results of gentamicin sensitivity?

We thank you for your comment. The use of gentamicin 120 µg was performed in order to detect high level aminoglycoside resistance (HLAR). In fact, this resistance is usually evaluated in Enterococcus species, however, there are also a few reports of HLAR in Enterobacter cloacae and Klebsiella pneumoniae. For that reason, we also evaluated HLAR in our collection.

  1. L297-299: “Then, inhibition zones were measured and scored according to CLSI guidelines as susceptible, intermediate, or resistant [48, 74]. If the authors used CLSI guidelines so, why they cited the EUCAST guidelines?

We appreciate your question. We have used the CLSI guidelines to determine the susceptibility profile of the isolates, as described in the methods section, since CLSI has specific breakpoints for isolates of veterinary origin. We did not use CLSI for the modified double disk synergy test, since it does not have breakpoints for this test. As such, the EUCAST Guidelines were used to perform the modified double disk synergy test.

  1. One of the most important mechanisms of antimicrobial resistance in Enterobacteriaceae is the enzymatic inactivation of penicillins and cephalosporins by means of plasmid-mediated extended-spectrum β-lactamases (ESBLs), such as the TEM-, SHV-, or cefotaxime (CTX)-M-group enzymes. Why do the authors use molecular tools to identify their isolates?

We understand your question and agree that molecular tools would allow the molecular characterization of the isolates, however, as previously stated, biochemical tests for bacteria identification are more economical, and we have no financial support to perform molecular identification. We would like, in the future, to perform an extensive molecular evaluation of this collection, and we are alert for project calls.

  1. L304: “Phenotypic evaluation of isolates’ virulence profile,” the authors determine some phenotypic properties of their isolates. But they do not identify the pathotypes of isolated E. coli and whether the Klebsiella pneumoniae are hypervirulent strains. Molecular detection of such virulence genes is essential.

We understand your comment, in fact, as we previous referred, we would like to perform the molecular characterization of this bacterial collection in the future, as the isolates are kept cryopreserved at -80ºC.

Results

  1. Based on the MDDST, the authors identified 48 ESBL-producing isolates from 29 dogs. In total, how many samples revealed more than isolate? Please clarify this point in the result section.

We thank you for your comment. This information was added in the results section (please see line 104).

  1. In supplementary materials, 6 isolates were recovered from the sample (ID 5). Four of them were ESBL-positive E. coli. Are they all the same strain? I think Multilocus sequence typing (MLST) is required to differentiate these isolates.

We agree with your comment. Although MLST would help to draw a final conclusion on the isolates’ clonality, as they present different antimicrobial susceptibility and virulence profiles we may infer that they do not correspond to the same strain.

Discussion:

  1. Please add your study limitation and a final conclusion.  

We agree with your suggestion and included information regarding the limitations of the study in the discussion section as well as a final conclusion, as you may see in lines 260-270 and 404-419.

Best regards,

Eva Cunha

Reviewer 3 Report

Remarks to the authors:

1. Please define abbreviations the first time they appear in the manuscript.

2. It would be beneficial to the manuscript if the authors included a brief section on the main hydrolytic properties of extended-spectrum -lactamases.

3. Please highlight the main significance and public health threat of human infections with ESBL.

4. What are the most common occurrences and epidemiology of ESBL acquired resistance?

5. what are the genes involved in public health concerns within this study?

6. What is the route of transmission of ESBL and the resistance genes to humans by consumption or handling of food-related environments?

7. what are the phenotypic and genotypic methods currently used for the detection of ESBL bacterial strains?

8. What are the risk factors contributing to the occurrence, emergence, and spread of ESBL?

9. What are the possible control options to reduce the public health risk caused by ESBL?

10. What are the primary controls for the selection and spread of extended-spectrum -lactamases?

Author Response

Dear Reviewer,

We would like to thank you for the comments and the discussion points presented. We agree that they will improve the quality and understanding of the manuscript. Please see below the response to each of your question. Also, all changes performed in the revised version of the manuscript were highlighted in yellow.

  1. Please define abbreviations the first time they appear in the manuscript.

We thank you for the comment. We have reviewed all manuscript in order to correct that.

  1. It would be beneficial to the manuscript if the authors included a brief section on the main hydrolytic properties of extended-spectrum -lactamases.

We agree with your suggestion and included new information regarding hydrolytic properties of extended-spectrum -lactamases (see Lines 39-45).

  1. Please highlight the main significance and public health threat of human infections with ESBL.

What are the most common occurrences and epidemiology of ESBL acquired resistance? what are the genes involved in public health concerns within this study? What is the route of transmission of ESBL and the resistance genes to humans by consumption or handling of food-related environments? what are the phenotypic and genotypic methods currently used for the detection of ESBL bacterial strains? What are the risk factors contributing to the occurrence, emergence, and spread of ESBL? What are the possible control options to reduce the public health risk caused by ESBL? What are the primary controls for the selection and spread of extended-spectrum -lactamases?

We understand your comments. We performed several changes all over the manuscript in accordance with your questions. In the introduction section we added information regarding the impact of ESBL in humans and animals (lines 61-74), the ESBL genes more relevant to public health (Lines 39-45), the transmission pathways of ESBL (lines 74-77) and the risk factors associated to the dissemination of ESBL (Lines 59-76). Then, in the conclusion section, we included some suggestions to reduce the public health risk associated to ESBL (lines 413-419).

Best regards,

Eva Cunha

Round 2

Reviewer 2 Report

The authors addressed most comments.